# Understanding how individualised physiotherapy or advice altered different elements of disability for people with low back pain using network analysis

**Bernard X. W. Liew**[1]*, **Jon J. Ford**[2], **Giovanni Briganti**[3], **Andrew J. Hahne**[2]

**1** School of Sport, Rehabilitation and Exercise Sciences, University of Essex, Colchester, Essex, United Kingdom, **2** Discipline of Physiotherapy, School of Allied Health, Human Services & Sport, La Trobe University, Melbourne Australia, **3** Department of Psychology, Harvard University, Cambridge, Massachusetts United States of America

\* bl19622@essex.ac.uk, liew_xwb@hotmail.com

## Abstract

**Data Availability Statement:** All codes and results can be found on Zenodo (DOI: 10.5281/zenodo. 5902763).

### Purpose

The Oswestry Disability Index (ODI) is a common aggregate measure of disability for people with Low Back Pain (LBP). Scores on individual items and the relationship between items of the ODI may help understand the complexity of low back disorders and their response to treatment. In this study, we present a network analysis to explore how individualised physiotherapy or advice might influence individual items of the ODI, and the relationship between those items, at different time points for people with LBP.

### Methods

Data from a randomised controlled trial (n = 300) comparing individualised physiotherapy versus advice for low back pain were used. A network analysis was performed at baseline, 5, 10, 26 and 52 weeks, with the 10 items of the Oswestry Disability Index modelled as continuous variables and treatment group (Individualised Physiotherapy or Advice) modelled as a dichotomous variable. A Mixed Graphical Model was used to estimate associations between variables in the network, while centrality indices (Strength, Closeness and Betweenness) were calculated to determine the importance of each variable.

### Results

Individualised Physiotherapy was directly related to lower Sleep and Pain scores at all follow-up time points relative to advice, as well as a lower Standing score at 10-weeks, and higher Lifting and Travelling scores at 5-weeks. The strongest associations in the network were between Sitting and Travelling at weeks 5 and 26, between Walking and Standing at week 10, and between Sitting and Standing scores at week 52. ODI items with the highest centrality measures were consistently found to be Pain, Work and Social Life.

**Funding:** LifeCare Health provided facilities, personnel and resources to allow treatment of trial participants free of charge in the STOPS Trial.

**Competing interests:** The authors have declared that no competing interests exist.

## Conclusion

This study represents the first to understand how individualised physiotherapy or advice differentially altered disability in people with LBP. Individualised Physiotherapy directly reduced Pain and Sleep more effectively than advice, which in turn may have facilitated improvements in other disability items. Through their high centrality measures, Pain may be considered as a candidate therapeutic target for optimising LBP management, while Work and Socialising may need to be addressed via intermediary improvements in lifting, standing, walking, travelling or sleep. Slower (5-week follow-up) improvements in Lifting and Travelling as an intended element of the Individualised Physiotherapy approach did not negatively impact any longer-term outcomes.

## Trials registration

ACTRN12609000834257.

## Introduction

Low Back Pain (LBP) is the leading cause of years lived with disability globally, with a point prevalence of 7.5% in 2017 [1]. In Western countries, the socio-economic cost of LBP has been estimated to be 1–2% of the gross national product [2, 3], particularly among individuals whose symptoms persist longer than six weeks (for this paper called post-acute LBP) [4]. The greater cost associated with post-acute LBP is unsurprising given that the rates of improvement in pain and disability plateau after six weeks from symptom onset [5]. It has been reported that between 28%-79% of participants reported incomplete recovery or had recurrent symptoms one year from study inception [5, 6]. Given the significant burden of disease associated with LBP, research into the treatment, prevention, and prognosis of this complex disorder has flourished over the past 30 years [5, 7, 8].

A primary outcome measure used in LBP research is the Oswestry Disability Index (ODI) for measuring the impact of LBP on activities of daily living [9]. The ODI is composed of 10 items and the aggregate score indicates the overall disability level attributable to LBP [9, 10]. ODI has demonstrated good internal consistency [11], intrinsic validity [12], test-retest reliability [13], and responsiveness [13]. A fundamental theoretical construct underpinning the contemporary use and interpretation of the ODI is known as the "reflective model" (RM) [14]. Put simply, observed item responses on the ODI are determined by a latent trait—disability. The principal advantage of using the aggregate score, over individual item scores, is that it makes it easier and more streamlined for statistical modelling in epidemiological and clinical research.

The reflective model interpretation of ODI total scores does have disadvantages. Firstly, two individuals could have identical ODI aggregate scores but with different item responses. Understanding what precisely is being affected in people with LBP is required for providing individualised treatment. Second, the relationship between different items of the ODI, a feature not captured when using only an aggregate score, maybe just as important as individual item responses in providing a holistic understanding of the functional limitations in individuals with LBP [15]. This would mean that simultaneous changes to the responses of multiple items, thereby influencing their relationship, may be important in determining recovery in individuals with LBP. In addition, understanding the relationship between different items may

be important given that changes to one item may "trigger" changes to other items (i.e. be correlated). For example, an individual who has difficulty standing may also have difficulty walking potentially due to the biomechanical loading and motor control similarities between these activities. Despite the stated disadvantages, using the aggregate score is still recommended in research focused solely on assessing the impact of LBP on overall disability.

A common critique of clinical intervention trials is that they fail to consider the complexity and multifactorial nature of conditions such as LBP [16, 17]. Qualitative studies have supported the notion that disability in people with LBP is a dynamic and complex construct [18–20]. A previous qualitative study proposed that perceptions of recovery in individuals with LBP may be best explained by an "interactive model", whereby symptoms, function and quality of life all interact to influence a person's perceived recovery [18]. A quantitative method to measure such an "interactive model" in LBP, and how such complex associations can be understood within the context of a clinical intervention, is network analysis [21]. In network analysis of the ODI for example, individual ODI items would be treated as *nodes*, and a network model would conceptualize LBP disability as a set of mutually interacting associations between these *nodes*. Associations between two nodes in a network are connected by an "edge" and reflect the magnitude of the relationship after statistically controlling for all other nodes in the network model [22]. Statistically, the association between two variables calculated in network analysis is analogous to the beta coefficient in a traditional multiple linear regression model, where one variable is the outcome and all the remaining variables are the predictors [22].

In contrast to network analysis, structural equations modelling (SEM) is a more common statistical technique used in spinal pain research to understand how different interventions alter relationships between multiple variables [23, 24]. Network analysis and SEM represent two alternate ways of describing the same variance-covariance structure of the modelled variables [25]. One key difference between the two approaches is that network analysis focuses on structural learning from the data (i.e. what variables are associated with each other), while SEM requires a fixed hypothesis to be tested with the data. In other words, network analysis focuses on hypothesis generation while SEM focuses on hypothesis confirmation. A second difference between SEM and network analysis is that SEM focuses on directional relationships whilst network analysis focuses on undirected (reciprocal) relationships. For example, it is known that poor sleep quality is associated with greater pain experience but greater pain can result in poor sleep [26], a reciprocal relationship that would be best suited to modelling via network analysis.

Given that there is little prior knowledge about the relationships between different elements of disability measured on the ODI, no prior knowledge on the impact of interventions on individual ODI items, and the plausibility that items on the ODI could be reciprocally related, network analysis represents a more appropriate technique than SEM for exploring how treatments influence individual items of the ODI. Network analysis has only recently been used in pain research [27, 28]; but has already been used substantially to investigate general psychopathologies [29–31]. To our knowledge, no studies in LBP research have used network analysis to understand how different treatment approaches influence different elements of disability as measured via individual items of the ODI.

To explore the impact of different interventions on various elements of disability via network analysis, data from a large randomised controlled trial comparing distinctive treatment approaches and measuring outcomes on the ODI across multiple timepoints is required. The Specific Treatment of Problems of the Spine (STOPS) trial was a large (n = 300) RCT concluding that individualised physiotherapy was more effective than advice for people with LBP in improving disability across a 12-month follow-up [32]. This dataset, with its high rate of

follow-up at all time points, was well suited to exploratory network analysis. The current study aimed to explore how the different treatments used in the STOPS trial influence individual items of the ODI at different follow-up stages. Such an approach has been previously used to understand the mechanisms of how cognitive-behavioural therapy positively influenced insomnia and depression [33]. Given that the STOPS trial reported a significantly greater improvement in overall disability with individualized physiotherapy compared to advice [32], we hypothesised that the variable of treatment group would positively influence several items of the ODI. As an exploratory study, however, we did not impose any prior assumptions on the direction or magnitude of interactions between any items of the ODI.

## Methods

### Study design

A network analysis was undertaken using data from the STOPS randomised controlled trial [32, 34]. The trial has received ethical approval from the La Trobe University Human Ethics Committee and has been registered with the Australian New Zealand Clinical Trials Registry (#12609000834257). Written informed consent was provided by participants prior to their inclusion. The network analysis explored relationships between individual items of the ODI, as well as the influence of treatment type (individualised physiotherapy versus advice) on those items, at multiple time points over a 12-month period.

### Participants

Participants were eligible for the trial if they: had a primary complaint of LBP (pain between the inferior costal margin and inferior gluteal fold), with or without referred leg pain, between 6 weeks and 6 months duration, were aged 18–65 years, spoke English, and belonged to one of five low back disorder subgroups (disc herniation with associated radiculopathy, reducible discogenic pain, non-reducible discogenic pain, zygapophyseal joint dysfunction, and multifactorial persistent pain) [32]. Exclusion criteria were: the presence of a compensation claim, serious pathology (active cancer, cauda equine syndrome, foot drop making walking unsafe), pregnancy or childbirth within the last 6 months, history of lumbar spine surgery, spinal injections in the past six weeks, pain intensity < 2/10 (on a 0–10 numerical rating scale) or minimal activity limitation. All selection criteria, including the diagnosis of LBP, were confirmed by a physiotherapist after an initial 60-minute assessment before entry to the trial.

### Randomisation

Eligible participants were randomised into one of two treatment groups via a computer-generated randomisation sequence; Individualised physiotherapy or advice. Allocation was concealed using an offsite randomisation service that allocated participants to treatment groups.

### Interventions (10 weeks)

Participants were randomised via an offsite randomisation service to either individualised physiotherapy (n = 156, 76 female and 80 male) or guideline-based advice (n = 144, 71 female and 73 male). Treatment was administered over 10-weeks. Participants were discharged to self-management. Outcomes were measured at baseline, and at 5, 10. 26 and 52-week follow-up.

**Guideline based advice.** Guideline-based advice comprised 2 x 30-minute sessions with a physiotherapist over 10 weeks based on the approach described by Indahl [35]. The first session was delivered shortly after randomisation, and the second approximately 4–5 weeks later.

Advice included an explanation of the hypothesized pain source, reassurance regarding a favourable prognosis, advice to remain active, and instruction regarding appropriate lifting technique [34].

**Individualised physiotherapy.** Individualised physiotherapy comprised 10 x 30-minute physiotherapy sessions over 10-weeks. Sessions were typically spaced weekly, although therapists had the option to deliver treatment more frequently in the first 2–3 weeks and less frequently in the last 2–3 weeks if clinically indicated. Physiotherapy was individualised firstly based on five subgroups, with further individualisation achieved within subgroups based on each participant's presenting barriers to recovery. Available treatment components included pathoanatomical or neurophysiological information, education, self-management strategies (posture, pacing, pain management, sleep management, relaxation strategies), inflammatory management strategies, exercise (specific muscle activation, goal-oriented graded activity/exercise), manual therapy (zygapophyseal joint dysfunction subgroup only), directional preference management (reducible discogenic pain subgroup only) and cognitive-behavioural strategies. Full details of the treatment protocols have been published previously [34, 36–39].

## Data collection

Self-administered questionnaires containing the ODI among other outcomes were posted to participants at each time point. Non-respondents were followed up directly and via alternative provided contacts if necessary.

## Approach to network analysis

**Software and packages.** The data set was analysed with the R software (version 3.6.0, available at https://www.r-project.org) [40]. Several packages were used to carry out the analyses, including qgraph [41], and mgm [42] for network estimation, and bootnet [43] for stability analysis. All codes and results can be found on the public code hosting platform GitHub (doi: 10.5281/zenodo.5902763).

**Variables included in network analysis.** A network structure is composed of nodes (variables influencing each other) and edges (connections or associations between nodes). In our study, the 10 items of a modified version of the ODI (that replaces the original "sex life" item with a "work/housework" item) were used as nodes and were included in the network model as continuous variables [9, 44], whilst treatment "group" was included as a dichotomous variable (coded as 0: advice, 1: individualised physiotherapy) (see Table 1 for abbreviated version;

**Table 1. Items and scorings of the Oswestry Disability Index.**

| Node | Variable | Least disabled (0) | Most disabled (5) |
|------|----------|--------------------|--------------------|
| Q1 | Pain intensity | I have no pain at the moment | The pain is the worst imaginable at the moment |
| Q2 | Personal care (washing, dressing, etc.) | I can look after myself normally without causing extra pain | I do not get dressed, wash with difficulty and stay in bed |
| Q3 | Lifting | I can lift heavy weights without extra pain | I cannot lift or carry anything |
| Q4 | Walking | Pain does not prevent me walking any distance | I am in bed most of the time |
| Q5 | Sitting | I can sit in any chair as long as I like | Pain prevents me from sitting at all |
| Q6 | Standing | I can stand as long as I want without extra pain | Pain prevents me from standing at all |
| Q7 | Sleeping | My sleep is never disturbed by pain | Pain prevents me from sleeping at all |
| Q8 | Social life | My social life is normal and gives me no extra pain | I have no social life because of pain |
| Q9 | Traveling | I can travel anywhere without pain | Pain prevents me from traveling except to receive treatment |
| Q10 | Work/Housework | My normal housework/work activities do not cause pain | Pain prevents me from performing any housework/work duties. |

full ODI scale is reported in the S1 File). The replacement of the "sex life" item with a "work/housework" item was previously undertaken to overcome frequent missing responses to the original sex life question [45]. This modified version has been shown to not adversely affect the construct validity of the ODI [44]. We treated the items of the ODI as continuous variables and applied a nonparanormal transformation to ensure that these ten variables were multivariate normally distributed [46]. Higher ODI aggregate and item scores represent greater disability.

Edges represent the existence of an association between two nodes, conditioned on all other nodes. Each edge in the network represents either a positive regularized association (blue edges) or a negative regularized association (red edges). Given their continuous nature, associations between items of the ODI reflect partial correlation coefficients, analogous to regression beta coefficients. Given that "group" was modelled with the advice group coded as 0 and the individualised physiotherapy group as 1, red edges (negative associations) between the "group" node and an ODI item would indicate that the individualised physiotherapy group had lower scores on that ODI item than the advice group [33]. If the two treatment groups had a similar influence on an ODI item score after controlling for all other items, an edge will not be present between the group and that ODI item. The thickness and colour saturation of an edge denotes its weight (the strength of the association between two nodes).

**Network estimation.** A Mixed Graphical Model was used to estimate the network [42]. Networks were estimated on complete datasets (n = 300, 283, 273, 271, 264) at each time point (baseline, weeks 5, 10, 26, 52), to understand the effects of different treatments on individual ODI items and their relationships at different time points. Least absolute shrinkage and selection operator (LASSO) regularization was used during modelling to elicit a sparse model. Compared to a saturated model, a sparse model is one with a comparatively fewer number of edges to explain the covariation structure of the data–with the benefit that the ensuing model becomes more interpretable [22].

**Node centrality.** Not all nodes in a network are equally important in determining the network structure [47]. Centrality indices provide a measure of a node's importance, and they are based on the pattern of connectivity of a node of interest with its surrounding nodes–with the ensuing information potentially useful for guiding future interventions [48].

In the present study, we calculated three centrality indices:

- Strength centrality is defined as the sum of the weights of the edges (in absolute value) incident to the node of interest [49, 50]. Clinically, a high Strength node represents a logical and efficient therapeutic target, because a change in the value of this node has a strong direct and quick (because of its strong direct connections) influence on other nodes within the network.

- Closeness centrality [49] is defined as the reciprocal of the sum of the length (inverse of the absolute value of edge's weight) of the shortest paths between a node of interest and all other nodes in the network. Clinically, a high Closeness node may represent a potentially good therapeutic target, because the effects of a change in the value of this node will spread more quickly throughout the network, via direct and indirect connections to other nodes.

- Betweenness centrality is defined as the number of times a node acts as a bridge along the shortest path between two other nodes. [49, 51]. Clinically, a high Betweenness node may suggest that the node represents a potential mediator since it acts as a bridge for "information flow" connecting different nodes, or even different clusters of nodes.

**Accuracy and stability.** We assessed the accuracy of the edge weights and the stability of three centrality indices using bootstrapping [43], which re-estimates the network parameters several times using a resampling technique. Accuracy and stability analyses are essential in network analysis studies to correctly interpret the results obtained. We bootstrapped using 2000 iterations, to generate 95% confidence intervals (CI) of all edge weights.

Given that LASSO regularization aims to estimate less important edges to be zero, the edge-weight bootstrapped CIs should not be interpreted using a null-hypothesis significance testing framework of a null relationship [43]. Instead, these edge weight CIs reflect the variability in estimated edge-weights and may be used to make a relative comparison of the different edge weights [43]. Whether the presence of an edge in the modelled network can be interpreted as such, is unaffected by large CIs as the LASSO already only keeps non-zero association edges in the model.

To gain an estimate of the variability of the three centrality indices, we applied the case-dropping subset bootstrap [43]. This procedure drops a percentage of participants, re-estimates the network and re-calculates the three centrality indices; producing a centrality-stability coefficient (CS-coefficient) that should not be lower than 0.25 and preferably above 0.5. CS reflects the maximum proportion of cases that can be dropped, such that with 95% probability the correlation between the centrality value of the bootstrapped sample versus that of the original data, would reach a certain value, taken to be a correlation magnitude of 0.7 presently. It is suggested that $CS_{cor = 0.7}$ should not be below 0.25 and better if $> 0.5$ [43].

## Results

The mean (standard deviation [SD]) of the variables (original scale) used in the network analysis can be found in the S1 File. Fig 1 shows the networks at baseline and at each of the four follow-up time points. Edge weights, variability and centrality indices values are reported in the manuscript graphically (Figs 2 and 3), but also as tabular text in the S1 File.

### Edge weights and variability

At baseline, Group was negatively related to Standing with a value of -0.09 (95%CI [-0.29 to 0]), whilst the edge with the greatest weight magnitude was between Sitting and Travelling with a value of 0.34 (95%CI [0.25 to 0.45]) (Fig 1). Given that ODI was administered before randomisation or any intervention at baseline, we interpreted the Group-Standing relationship as adjusted baseline differences in Standing response between the two groups that likely arose by chance given the randomised nature of the study.

The IP Group was directly associated with lower Sleep and Pain scores at all follow-up time points, with the strongest relation with Sleep occurring at week 5: -0.24 (95%CI [-0.42 to -0.07]), and the strongest relation with Pain at week 26: -0.15 (95%CI [-0.34 to 0]) (Figs 1 and 2). The IP Group was associated with higher Lifting and Travelling scores with values of 0.16 (95%CI [0 to 0.35]), and 0.08 (95%CI [0 to 0.27]), respectively, at week 5 only (Figs 1 and 2). At week 10, the IP Group was related to lower Standing scores with a value of -0.07 (95%CI [-0.32 to 0]) (Figs 1 and 2).

The edge with the greatest weight magnitude in the network was between Sitting and Travelling at week 5 with a value of 0.41 ([95%CI (0.3 to 0.5]); between Walking and Standing at week 10 with a value of 0.32 ([95%CI (0.21 to 0.45]; between Sitting and Travelling at week 26 with a value of 0.37 ([95%CI (0.24 to 0.48]); and between Sitting and Standing at week 52 with a value of 0.3 ([95%CI (0.18 to 0.41]) (Figs 1 and 2).

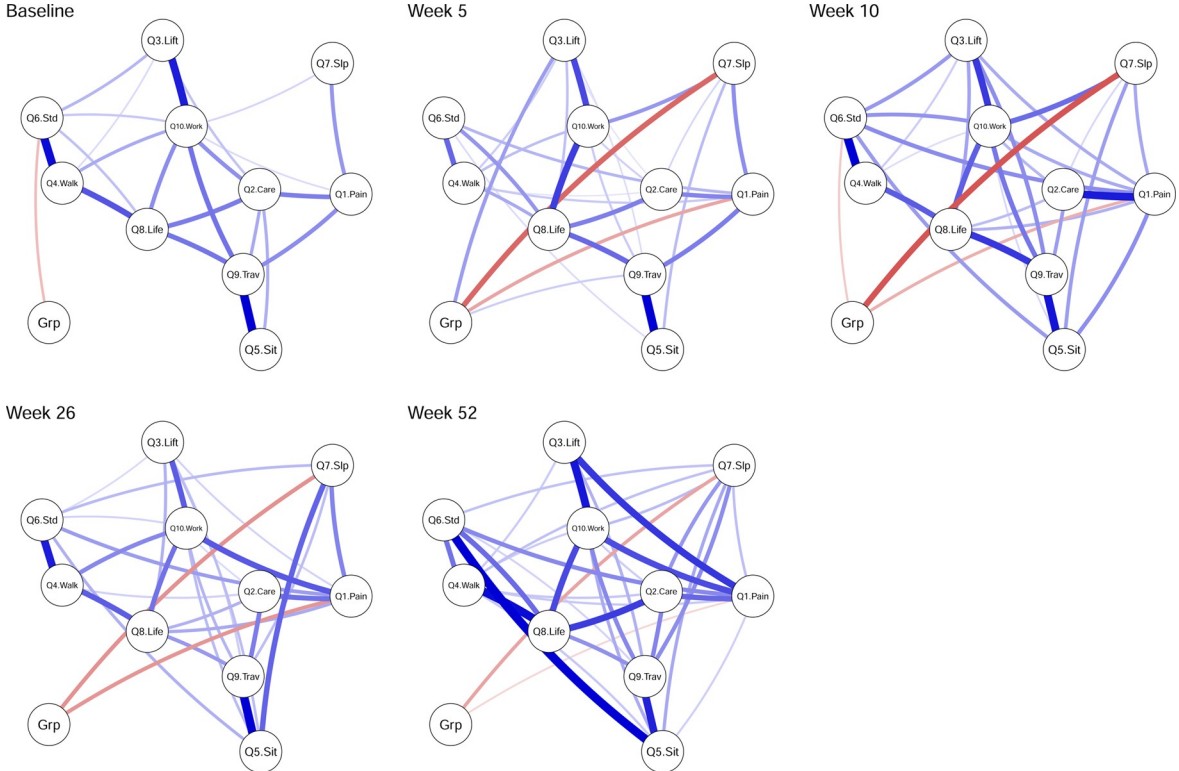

**Fig 1. Network analysis of the association between items of the Oswestry Disability Index and treatment group, at five follow-up time points.** Edges represent connections between two nodes and are interpreted as the existence of an association between two nodes, adjusted for all other nodes. Each edge in the network represents either positive regularized adjusted associations (blue edges) or negative regularized adjusted associations (red edges). The thickness and colour saturation of an edge denotes its weight (the strength of the association between two nodes). Abbreviation: Q1 –Pain Intensity, Q2 –Personal Care, Q3 –Lifting, Q4 –Walking, Q5 –Sitting, Q6 – Standing, Q7 –Sleeping, Q8 –Social life, Q9 Travelling, Q10 –Work/Housework.

## Centrality and variability

Across all three centrality measures, the two nodes with the greatest averaged value were Social life and Travelling at baseline, Social life and Work at week 5, Pain and Work at week 10, Pain and Work at week 26, and Social life and Work at week 52 (Fig 3). The stability of the centrality measures across all follow-up time points can be found in Table 2, which suggests that the centrality measure of Strength was most stable, i.e. $CS_{cor = 0.7} > 0.25$, across all time points (Fig 4).

## Discussion

Disability is a complex construct that emerges as a consequence of the simultaneous influence of many biopsychosocial factors. To capture the complex mechanisms of different physiotherapy approaches for low back pain, we used network analysis applied to individual items of the ODI. This is the first study to investigate how different physiotherapy approaches influence individual items of the ODI, and the relationships between those items. Three important findings are revealed by the network analysis. First, IP was better than advice at improving Pain intensity and Sleep quality at all follow-up time points of 5, 10, 26, and 52 weeks, which in turn may have facilitated positive changes in other items of the ODI. Second, Pain, Social life, and Work demonstrated the highest centrality measures in the network. As such, Pain may be considered as a candidate therapeutic target for optimising LBP management, while Work and Socialising may be best influenced by targeting functional components that impact these

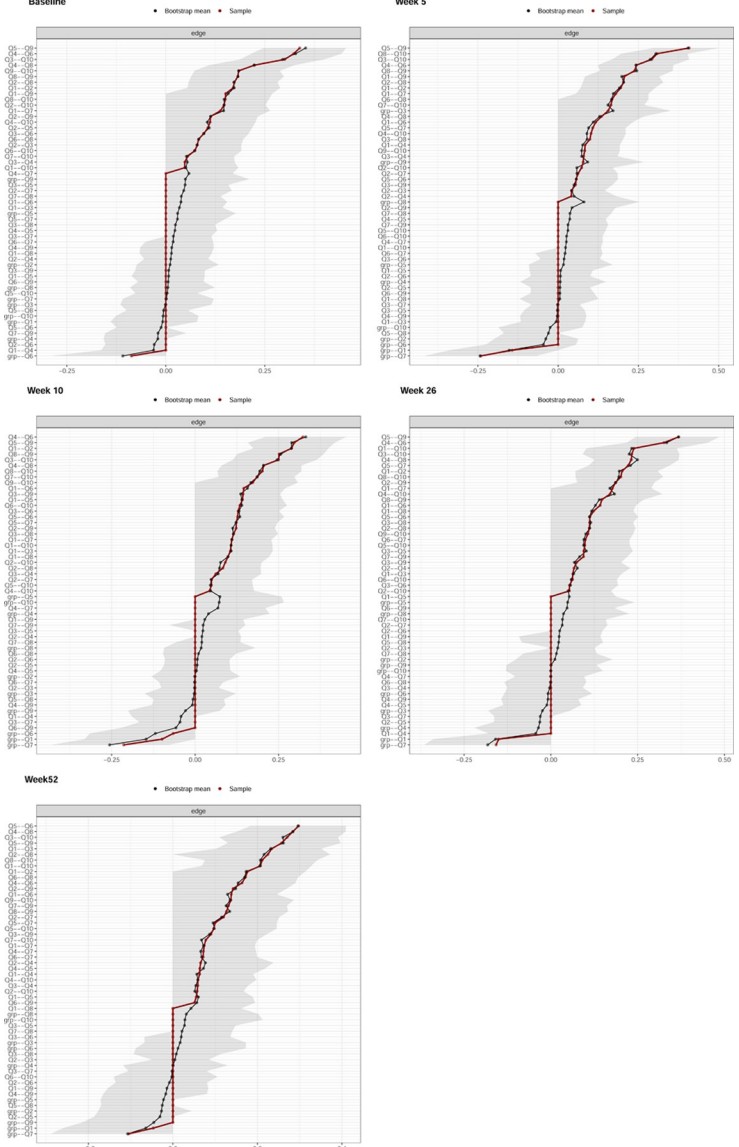

**Fig 2. Bootstrapped 95% quantile confidence interval of the estimated edge weights of the network at all follow-up time points.** "Bootstrap mean" reflects the average magnitude of edge weights across the bootstrapped samples. "Sample" reflects the magnitude of edge weights of the original network built on the entire input dataset. Abbreviation: Q1 –Pain Intensity, Q2 –Personal Care, Q3 –Lifting, Q4 –Walking, Q5 –Sitting, Q6 –Standing, Q7 –Sleeping, Q8 – Social life, Q9 Travelling, Q10 –Work/Housework.

activities such as lifting, standing, walking, travelling or sleep quality. Third, Lifting and Travelling improved less with IP than advice in the short term (week 5), likely due to the different aims of the two treatment programs, but this was not to the detriment of longer-term outcomes.

Participants who received IP achieved a greater improvement in Sleep at all follow-up time points compared to those who received advice. Our study found that Sleep lay on the path between Group and Pain, but Sleep also had a significant direct link to Group, suggesting that IP positively influenced Sleep by both direct and indirect pathways. Specific management of sleep impairment—which included management of inflammation, sleep posture and sleep

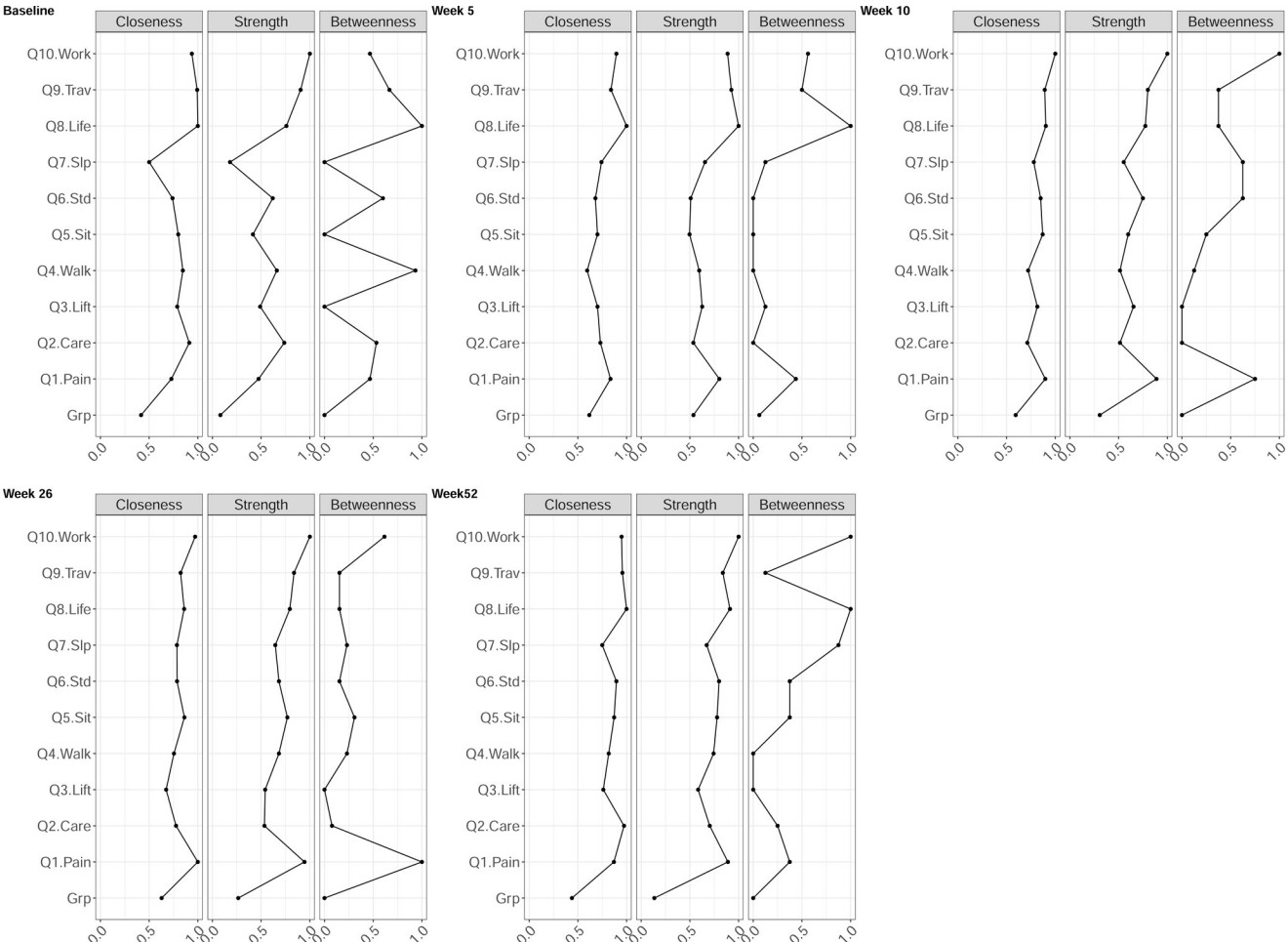

**Fig 3. Centrality measures of Closeness, Strength, and Betweenness of each node in the network at all follow-up time points.** Centrality value of 1 indicates maximal importance, and 0 indicates no importance. Abbreviation: Q1 –Pain Intensity, Q2 –Personal Care, Q3 –Lifting, Q4 –Walking, Q5 – Sitting, Q6 –Standing, Q7 –Sleeping, Q8 –Social life, Q9 Travelling, Q10 –Work/Housework.

hygiene strategies, was provided in the IP but not advice group [34]. This may explain the direct pathway between Group and Sleep, where IP was more effective than advice for improving Sleep at all follow-up time points. The indirect pathway between Group and Sleep involved the Sleep-Pain relationship. IP had a significantly better effect on improving pain intensity than advice, which in turn improved sleep. Given that a bidirectional relationship between sleep and pain is well established [26], IP potentially provides two pathways for treating pain or sleep dysfunction: either by directly targeting the outcome of interest or by targeting the other variable in the sleep-pain relationship to influence the other. Our findings provide multiple options for clinicians hoping to improve particular variables by considering synergistic bidirectional relationships between multiple variables. This is in contrast to some previous studies that only consider unidirectional relationships, such as when assuming a one-way relationship between sleep and pain [52, 53].

It has been suggested that the nodes which optimally span the network, evidenced by high centrality measures, may be the most important variables to target in an intervention program [54]. In the present study, Pain, Social life and Work were the most important nodes at follow-up in the network. Single (eg. cognitive behavioural therapy) [55] and multi-disciplinary

**Table 2. Centrality stability (CS) at each follow-up time points.**

| time | measure | CS |
|---:|---|---:|
| 0 | betweenness | 0.00 |
| 0 | closeness | 0.00 |
| 0 | strength | 0.59 |
| 5 | betweenness | 0.21 |
| 5 | closeness | 0.21 |
| 5 | strength | 0.28 |
| 10 | betweenness | 0.05 |
| 10 | closeness | 0.13 |
| 10 | strength | 0.28 |
| 26 | betweenness | 0.05 |
| 26 | closeness | 0.28 |
| 26 | strength | 0.44 |
| 52 | betweenness | 0.00 |
| 52 | closeness | 0.00 |
| 52 | strength | 0.59 |

treatment programs [56] are effective for improving pain, function and psychosocial outcomes in post-acute LBP. In the present study, Pain was directly reduced by IP more than by advice, and the high centrality of Pain in the network suggests that it may have subsequently improved multiple connected variables. This is consistent with findings reported in the STOPS trial [32], where numerous superior outcomes were attributable to the IP intervention (including pain, disability and psychosocial factors including work). Although a preliminary finding, the current network analysis suggests that targeting pain in an IP treatment program may be an efficient way to achieve positive outcomes on several other biopsychosocial variables.

Previous studies have demonstrated that LBP can have a profound negative impact on an individual's social life [57, 58] and work [58, 59]. The high centrality measures for Social Life and Work found in this study are therefore important findings that could have various interpretations. Firstly, these variables may be candidate therapeutic targets most likely to influence many other items of the ODI and overall disability. In that case treatment approaches (including IP) could potentially become more efficient by aiming to directly improve an individual's capacity to engage in meaningful social activities and work (as the variables with high centrality ratings) rather than impacting those variables through indirect or intermediary pathways. Alternatively, it may be difficult to directly influence complex multifactorial outcomes such as Social life and Work without intermediary mechanisms. In that case, due to bidirectionality in the network, improvements in Social Life and Work could potentially be achieved by targeting one or more of several direct connections in the network (such as lifting, standing, walking, travelling or sleeping). It is plausible that these connected variables could individually or cumulatively impact a patient's ability to socialise or work, which may explain their high degree of centrality in the network. A strength of the network analysis is the large number of potential treatment targets that could, directly and indirectly, improve an outcome of interest. Whether a network informed precision treatment approach could assist clinicians to better target their individualised or stratified treatment needs to be validated in future studies.

A potentially surprising finding of the present study was that Group was positively associated with Lifting and Travelling at week 5, indicating that IP improved these items less than the advice group at this early time point. This relationship was not due to an artefact of our analysis (e.g. adjusting for a common effect between two nodes [22]), as it can be corroborated

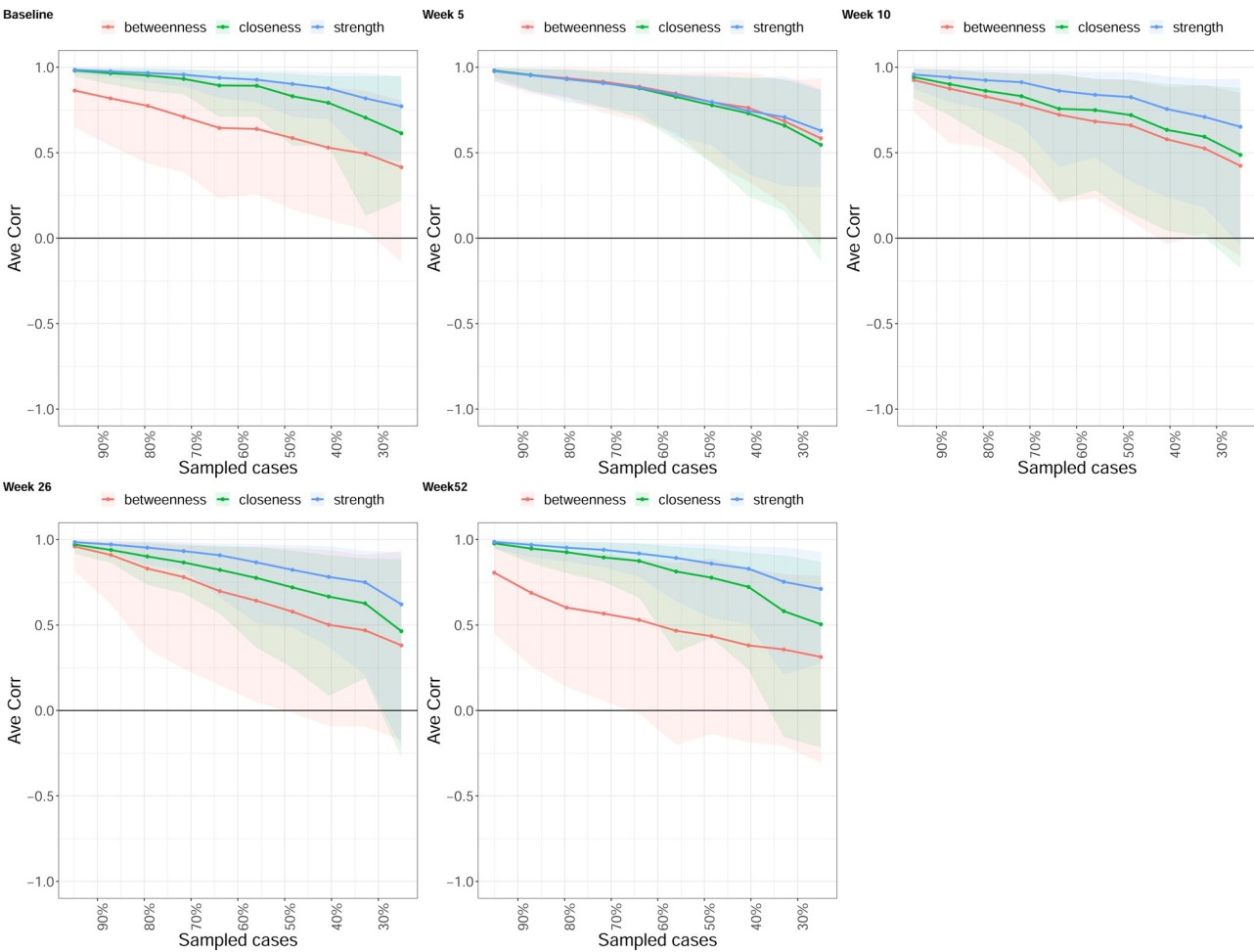

**Fig 4. Average correlations between centrality indices of networks sampled with persons dropped and networks built on the entire input dataset, at all follow-up time points.** Lines indicate the means and areas indicate the range from the 2.5th quantile to the 97.5th quantile.

by simple descriptive measures (S1 File). This finding is consistent with the intent of the respective IP and advice treatment protocols. The aim of advice was to facilitate a quick return to all activities. By contrast, the emphasis of the IP intervention in the first 5 weeks for many patients was to minimise provocative activities such as lifting and prolonged sitting/driving while underlying pathophysiological processes (such as inflammation, discogenic pathology, and suboptimal motor control) were addressed, before a graded return to all activities was facilitated over the subsequent weeks. Some would argue that restricting activities such as lifting or sitting/driving for people with LBP is commonly unnecessary or counterproductive [60]. However, our study suggests that temporarily restricting these potentially provocative activities for the intervertebral disc could be justified. Despite limiting lifting and sitting/driving for the first five weeks in many participants in the IP group, we found that neither Lifting or Travelling (or any other ODI item) were associated with higher scores in the IP group at the 10–52 week follow-ups, and the overall outcomes for pain (5, 10 & 26 weeks) and function (10, 26 and 52 weeks) favoured the IP group in the RCT [32]. This finding confirms our rationale for the current study that looking at relationships between individual ODI items (rather than only looking at the overall score) may help understand how different interventions impacted individual elements of disability at particular time points.

Given the novelty of network analysis, it may be prudent to clarify the limitations of inferences drawn from them. Conditional independence relationships, as encoded by the edge weights in the networks, cannot be a source of confirmatory causal inference, but may provide indicative potential causal pathways [22, 61]. For example, if all relevant variables are modelled in a network, an observed adjusted association between variables X and Y would only be possible if, either X causes Y, Y causes X, X and Y exhibits a bidirectional relationship, or X and Y have a common effect [22, 61]. Hence, network analysis may be conceptualized as a highly exploratory hypothesis-generating technique, indicative of *potential* causal effects. Another limitation is that no statistical comparisons between networks at different time points could be undertaken. This meant that whether alterations across time in network structure, edge weights, and node centrality, were statistically different were not quantified. Readers could make a qualitative judgment as to whether two networks across time differ, but caution should be exercised in making any substantive conclusions about their differences. Although network comparison tests via permutation are available [62], the current implementation of the software package is limited to the analysis of either a network containing only continuous variables, or a network containing only binary variables.

## Conclusions

This study represents the first to understand how individualised physiotherapy or advice differentially altered disability in people with LBP. IP directly reduced Pain and Sleep more effectively than advice, which in turn may have facilitated improvements in other ODI items. Through their high centrality measures, Pain may be considered as a candidate therapeutic target for optimising LBP management, while Work and Socialising may need to be addressed via intermediary improvements in lifting, standing, walking, travelling or sleep. Slower (5-week follow-up) improvements in Lifting and Travelling as an intended element of the IP group approach did not negatively influence any longer-term outcomes.

## Supporting information

**S1 File.**
(ZIP)

## Acknowledgments

The authors wish to acknowledge the trial physiotherapists who volunteered to treat participants in this trial free of charge.

## Author Contributions

**Conceptualization:** Bernard X. W. Liew, Jon J. Ford, Andrew J. Hahne.

**Data curation:** Bernard X. W. Liew, Jon J. Ford, Andrew J. Hahne.

**Formal analysis:** Bernard X. W. Liew, Giovanni Briganti.

**Funding acquisition:** Jon J. Ford, Andrew J. Hahne.

**Investigation:** Jon J. Ford, Andrew J. Hahne.

**Methodology:** Bernard X. W. Liew, Giovanni Briganti.

**Project administration:** Jon J. Ford, Andrew J. Hahne.

**Software:** Giovanni Briganti.

**Supervision:** Jon J. Ford, Andrew J. Hahne.

**Validation:** Jon J. Ford, Giovanni Briganti, Andrew J. Hahne.

**Visualization:** Bernard X. W. Liew, Giovanni Briganti.

**Writing – original draft:** Bernard X. W. Liew, Andrew J. Hahne.

**Writing – review & editing:** Bernard X. W. Liew, Jon J. Ford, Giovanni Briganti, Andrew J. Hahne.

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
