## [Decision Letter · Decision Letter 0]

21 May 2021

PONE-D-21-12519

Understanding how individualised physiotherapy or advice altered different elements of disability for people with low back pain using network analysis.

PLOS ONE

Dear Dr. Liew,

Thank you for submitting your manuscript to PLOS ONE. After careful consideration, we feel that it has merit but does not fully meet PLOS ONE’s publication criteria as it currently stands. Therefore, we invite you to submit a revised version of the manuscript that addresses the points raised during the review process.

Thank you for submitting this interesting paper.

Please find below the feedback from 3 reviewers including one from a statistical background that highlight some key points that need to be addressed.

We look forward to receiving your revised manuscript.

Kind regards,

Alison Rushton

Academic Editor

PLOS ONE

Journal Requirements:

Reviewers' comments:

Reviewer's Responses to Questions

**Comments to the Author**

1. Is the manuscript technically sound, and do the data support the conclusions?

Reviewer #1: Yes

Reviewer #2: Yes

Reviewer #3: Yes

2. Has the statistical analysis been performed appropriately and rigorously? 

Reviewer #1: Yes

Reviewer #2: Yes

Reviewer #3: Yes

3. Have the authors made all data underlying the findings in their manuscript fully available?

Reviewer #1: Yes

Reviewer #2: Yes

Reviewer #3: Yes

4. Is the manuscript presented in an intelligible fashion and written in standard English?

Reviewer #1: Yes

Reviewer #2: Yes

Reviewer #3: Yes

5. Review Comments to the Author

Reviewer #1: 

I will focus on methods and reporting. Overall there is nothing wrong with the paper from a methodological point of view. I am not familiar with these methods so I had to look into them, but everything looks acceptable. It does seem like using something overly complex to examine something relatively straightforward. That is the authors' prerogative however, and I can accept it.

What perhaps needs to be strengthened is the "why", from a clinical point of view. "Dynamic relationships" is mentioned quite a few times, but it does not shed any light on the clinical importance and the "so what" of the paper. Some work is needed there.

From a technical point of view, I don't see what repeating the same analysis at each time point adds, and how it is possible to get such diverse results a few weeks apart. Isn't that evidence that the variability is so great that interpretation needs to be cautious? How do the authors explain that? Also, as I previously implied, my preference would be for a more established method (like structural equation modelling) that accounted for time and estimated all relationships in a single model. What does this approach offer that is better?

Reviewer #2: 

First of all, I would like to thank the authors for this very interesting article which tackles LBP from a rather original angle. The study uses data from a previous study (carried out by one of the authors of this study) to perform a network analysis. This analysis is based on the analysis of the links between the different items of a LBP evaluation questionnaire. The originality of this article is both an advantage and a disadvantage. Indeed no study has carried out such work previously, which is positive. However, few physiotherapists are familiar with network analysis, which makes it very difficult to understand the article, especially in the Materials and Methods part. This last remark leads me to think that this part should be reviewed partly (simplify or better illustrate with examples to understand what has been done) before being accepted.

About the abstract: nothing to declare about this part of the article.

introduction:

line 56: Low Back Pain (with capital letters for LBP)

line 63: about the first clinical consultation : what is the patient care path?

line 68: about ODI : as this is the core of the study, it would increase the robustness of the choice by adding more literature on ODI

line 75: when talking about the disadvantages of ODI, it should be explained how these disadvantages remain acceptable and do not compromise its use in the study

line 88 to 92:this is a very important point of the study, which deserves to be further developed

line 92 à 96:As mentioned previously, network analysis is almost unknown to physiotherapists, so it seems very important to me to spend more time explaining it in a simple way. this would make the article much more accessible.

line 96-97: you should rework the transition between these 2 paragraphs because it is too abrupt, there is no clear link. we do not understand why we go directly to STOPS.

line 99: is it [22] or [23] or both ?

The introduction ends abruptly without the research question, without a working hypothesis, without the objectives, etc. This should be added.

Materials and Methods:

This part is the most difficult to understand and I think you are aware of it. This part is however very complete and I have nothing to say about the methodology which seems to be very rigorous. Much effort has been made to make it clear, but in my opinion there are still elements that need to be clarified to facilitate the reader's understanding.

line 107 : how were patients diagnosed with LBP ?

line 113: how was the pain assessed at 2/10 (tool)

line 138 : is it a typo problem [29] [29] [29] ?

line 144-146:Why did you modify the ODI scale and replace the item on "sexual life" ? In my opinion, it should have been left for two reasons. First, this is an important question for LBP patients. And especially secondly because it modifies the scale that was originally validated with this item. It should be explained how changing the scale does not compromise its use in the study. Especially since the item "Work" which replaces it seems very important in the results ... it is in my opinion the first "problem" of the study.

the second problematic point about the study is that I do not understand how we can individualize the IP and advice group ... I reread the materials and methods several times, I looked at the figures but I cannot understand how you draw conclusions by saying that the IP is better on an item compared to the advice group.

This lack of understanding highlights the complexity of the study. This remark may surprise you but I think that the readers will not understand, by looking at the figures how you manage to differentiate the groups on the items.

It must be explained more clearly in the materials and method how the 2 groups will be differentiated. and in the discussion part, give examples to understand the difference between groups

Discussion:

The discussion is rather well structured and clear and I do not have any particular remarks. However, as I said, we do not understand the conclusions obtained with regard to the results.

References: put the numbers in square brackets? [1], etc.

In conclusion, I would say that this study is very interesting, complete, well constructed and above all original. I recommend its publication AFTER making changes that will improve the reader's understanding.

The 2 central points are:

- The problem of modifying the ODI scale on an item: explain how this does not call into question the validity of the scale.

- And above all: explain to the reader how to read the figures to highlight the differences between the 2 groups in the study. The use of example seems important to me since the target audience (physiotherapists) are not always clear about network analysis.

In any case, well done for this work.

Best regards

Reviewer #3: 

1. Define LBP

2. please Specify the conditions/disease which causes LBP in your inclusion criteria

3. why you have taken 18-65 age groups subjects.

4. mention gender distribution ( how many males and females)

5. mention tool used for measuring pain intensity

6. mention ODI psychometric properties

7. duration of protocol should be mention per day tretment duration/days per week/ total weeks/days

8. Randomisation procedure ( GRoup division is not mentioned clearly- Group A/ Group B-so on)

9. review your Methodology ( intervention and data collection)

10. check references format

11. table 1 is not easily understandable. make it in simple format

12. review references and their format.

13. make a table of results obtained at different intervals

6. PLOS authors have the option to publish the peer review history of their article (what does this mean?). If published, this will include your full peer review and any attached files.

Reviewer #1: No

Reviewer #2: **Yes: **Lucas Martinez

Reviewer #3: No

---

## [Author Response · Author response to Decision Letter 0]

21 Jun 2021

Please see the attached "response to reviewers" document below to more conveniently see our responses, which has proper formatting, citations, and figures.

Reviewer #1: 

I will focus on methods and reporting. Overall there is nothing wrong with the paper from a methodological point of view. I am not familiar with these methods so I had to look into them, but everything looks acceptable. It does seem like using something overly complex to examine something relatively straightforward. That is the authors' prerogative however, and I can accept it.

What perhaps needs to be strengthened is the "why", from a clinical point of view. "Dynamic relationships" is mentioned quite a few times, but it does not shed any light on the clinical importance and the "so what" of the paper. Some work is needed there.

Reply: We have emphasized in the Introduction how traditional analysis fails to capture the complexity inherent in the recovery process of individuals with LBP. We have also made a stronger emphasis on the clinical impact of the paper in the Introduction, Discussion, and Conclusion.

In L91:

A common critique of clinical intervention trials is that they fail to consider the complexity and multifactorial nature of conditions such as LBP [16, 17]. Qualitative studies have supported the notion that disability in people with LBP is a dynamic and complex construct [18-20]. 

L355:

Disability is a complex construct that emerges as a consequence of the simultaneous influence of many biopsychosocial factors. To capture the complex mechanisms of different physiotherapy approaches for low back pain, we used network analysis applied to individual items of the ODI. 

The main implications for clinical practice are summarized in the conclusion (L463): 

Pain may be considered as a candidate therapeutic target for optimising LBP management, while Work and Socialising may need to be addressed via intermediary improvements in lifting, standing, walking, travelling or sleep. 

From a technical point of view, I don't see what repeating the same analysis at each time point adds, and how it is possible to get such diverse results a few weeks apart. Isn't that evidence that the variability is so great that interpretation needs to be cautious? How do the authors explain that? Also, as I previously implied, my preference would be for a more established method (like structural equation modelling) that accounted for time and estimated all relationships in a single model. What does this approach offer that is better?

Reply: We thank the Reviewer for these important comments. There are three points made by the Reviewer, which we will address individually. 

First, the Reviewer correctly commented on the appropriateness of repeated network analysis. In the revised Introduction, we now cite a precedent study that has used a similar approach to us, albeit in a different context.

L137:

Such an approach has been previously used to understand the mechanisms of how cognitive-behavioural therapy positively influenced insomnia and depression [33]. 

When viewed within the lens of traditional statistics, the Reviewer’s comment is analogous to valid concerns sometimes expressed about performing repeated two-sample t-tests rather than a single repeated-measures ANOVA. Both approaches are valid, but the preference of a repeated-measures ANOVA is that it guards against false-positive findings. However, there is currently no approach to conduct a version of repeated-measures ANOVA in network analysis. Our rationale in analysing each timepoint was that it is plausible that treatment effects (or the maintenance of treatment effects after discharge) could vary at different time points. As it eventuated, a key study finding was facilitated by this decision whereby we observed a smaller improvement in travelling and lifting in the IP group at 5 weeks (consistent with the treatment approach to avoid these activities in the early stages) that did not carry any longer-term consequences on any outcomes for this group. 

L235: 

Networks were estimated on complete datasets (n = 300, 283, 273, 271, 264) at each time point (baseline, weeks 5, 10, 26, 52), to understand the effects of different treatments on individual ODI items and their relationships at different time points.

Second, the Reviewer states that our networks across time may be too variable for meaningful interpretation. At present, we cannot determine if our networks across time points significantly differ both in their structure and in the magnitude of associations, as the statistical tools to perform such comparisons are presently not available. We have mentioned this in the limitations of the original manuscript. We also performed and reported the results of stability analyses, to indicate to the readers how likely a particular association was to occur again in repeated experiments. In addition, our findings were relatively stable across time points, and any unique relationships at individual time points were able to be explained in the Discussion (eg. our finding of a lower improvement in Travelling and Lifting at 5 weeks in the IP group mentioned above). 

However, we agree with the Reviewer that readers should exercise caution in interpreting our results until such time where hypothesis testing of mixed-graphical models becomes available. We have added to the limitation section in the revised manuscript in L453:

Readers could make a qualitative judgment as to whether two networks across time differ, but caution should be exercised in making any substantive conclusions about their differences.

Third, in the revised Introduction (L107), we added a section on why network analysis is a more appropriate technique than SEM for the present study.

In contrast to network analysis, structural equations modelling (SEM) is a more common statistical technique used in spinal pain research to understand how different interventions alter relationships between multiple variables [23, 24]. Network analysis and SEM represent two alternate ways of describing the same variance-covariance structure of the modelled variables [25]. One key difference between the two approaches is that network analysis focuses on structural learning from the data (i.e. what variables are associated with each other), while SEM requires a fixed hypothesis to be tested with the data. In other words, network analysis focuses on hypothesis generation while SEM focuses on hypothesis confirmation. A second difference between SEM and network analysis is that SEM focuses on directional relationships whilst network analysis focuses on undirected (reciprocal) relationships. For example, it is known that poor sleep quality is associated with greater pain experience but greater pain can result in poor sleep [26], a reciprocal relationship that would be best suited to modelling via network analysis. 

Given that there is little prior knowledge about the relationships between different elements of disability measured on the ODI, no prior knowledge on the impact of interventions on individual ODI items, and the plausibility that items on the ODI could be reciprocally related, network analysis represents a more appropriate technique than SEM for exploring how treatments influence individual items of the ODI. Network analysis has only recently been used in pain research [27, 28]; but has already been used substantially to investigate general psychopathologies [29-31]. To our knowledge, no studies in LBP research have used network analysis to understand how different treatment approaches influence different elements of disability as measured via individual items of the ODI. 

 

Reviewer #2: 

First of all, I would like to thank the authors for this very interesting article which tackles LBP from a rather original angle. The study uses data from a previous study (carried out by one of the authors of this study) to perform a network analysis. This analysis is based on the analysis of the links between the different items of a LBP evaluation questionnaire. The originality of this article is both an advantage and a disadvantage. Indeed no study has carried out such work previously, which is positive. However, few physiotherapists are familiar with network analysis, which makes it very difficult to understand the article, especially in the Materials and Methods part. This last remark leads me to think that this part should be reviewed partly (simplify or better illustrate with examples to understand what has been done) before being accepted.

Reply: We thank the Reviewer for the positive comments about this paper, and will address all feedback provided below.

About the abstract: nothing to declare about this part of the article.

introduction:

line 26: Low Back Pain (with capital letters for LBP)

Reply: We have reworded this to Low Back Pain.

line 63: about the first clinical consultation : what is the patient care path?

Reply: We thank the Reviewer for this comment. The reviews cited in this sentence included a heterogeneous sample of studies recruiting from the community, general medical practices, and rheumatology; with no detailed specification of their clinical care path. As such, we have decided to reword this sentence in L61:

It has been reported that between 28%-79% of participants reported incomplete recovery or had recurrent symptoms one year from study inception [5, 6].

line 68: about ODI : as this is the core of the study, it would increase the robustness of the choice by adding more literature on ODI

Reply: We have added more information about the ODI, especially on its psychometric properties. This is found in L66.

A primary outcome measure used in LBP research is the Oswestry Disability Index (ODI) for measuring the impact of LBP on activities of daily living [9]. The ODI is composed of 10 items and the aggregate score indicates the overall disability level attributable to LBP [9, 10]. ODI has demonstrated good internal consistency [11],intrinsic validity [12], test-retest reliability [13], and responsiveness [13]. 

line 75: when talking about the disadvantages of ODI, it should be explained how these disadvantages remain acceptable and do not compromise its use in the study

Reply: We have added a sentence in L88.

Despite the stated disadvantages, using the aggregate score is still recommended in research focused solely on assessing the impact of LBP on overall disability.

line 88 to 92:this is a very important point of the study, which deserves to be further developed

Reply: We have further developed this paragraph in L91.

A common critique of clinical intervention trials is that they fail to consider the complexity and multifactorial nature of conditions such as LBP [16, 17]. Qualitative studies have supported the notion that disability in people with LBP is a dynamic and complex construct [18-20]. A previous qualitative study proposed that perceptions of recovery in individuals with LBP may be best explained by an “interactive model”, whereby symptoms, function and quality of life all interact to influence a person's perceived recovery. [18]. A quantitative method to measure such an “interactive model” in LBP, and how such complex associations can be understood within the context of a clinical intervention, is network analysis. [21]. In network analysis of the ODI for example, individual ODI items would be treated as nodes, and a network model would conceptualize LBP disability as a set of mutually interacting associations between these nodes. Associations between two nodes in a network are connected by an “edge” and reflect the magnitude of the relationship after statistically controlling for all other nodes in the network model [22]. Statistically, the association between two variables calculated in network analysis is analogous to the beta coefficient in a traditional multiple linear regression model, where one variable is the outcome and all the remaining variables are the predictors [22].

line 92 à 96:As mentioned previously, network analysis is almost unknown to physiotherapists, so it seems very important to me to spend more time explaining it in a simple way. this would make the article much more accessible.

Reply: We thank the Reviewer for these comments. We attempted to cater to a wide readership including clinicians, researchers, and statisticians who may be interested in our paper. However, we have now re-worked the paper to try and make it easier to follow for those who may be unfamiliar with network analysis (which will likely be a large proportion of the readership). We have addressed this in the response to the previous comment. In particular, we emphasize how the associations calculated between variables with network analysis are analogous to regression coefficients in a more traditional multiple linear regression model which most readers will be more familiar with.

L99: 

In network analysis of the ODI for example, individual ODI items would be treated as nodes, and a network model would conceptualize LBP disability as a set of mutually interacting associations between these nodes. Associations between two nodes in a network are connected by an “edge” and reflect the magnitude of the relationship after statistically controlling for all other nodes in the network model [22]. Statistically, the association between two variables calculated in network analysis is analogous to the beta coefficient in a traditional multiple linear regression model, where one variable is the outcome and all the remaining variables are the predictors [22]. 

line 96-97: you should rework the transition between these 2 paragraphs because it is too abrupt, there is no clear link. we do not understand why we go directly to STOPS.

Reply: We thank the Reviewer for this comment. We have modified the current paragraph to improve the transition leading on from the previous paragraph.

L129:

To explore the impact of different interventions on various elements of disability via network analysis, data from a large randomised controlled trial comparing distinctive treatment approaches and measuring outcomes on the ODI across multiple timepoints is required. The Specific Treatment of Problems of the Spine (STOPS) trial was a large (n=300) RCT concluding that individualised physiotherapy was more effective than advice for people with LBP in improving disability across a 12-month follow-up [32]. This dataset, with its high rate of follow-up at all time points, was well suited to exploratory network analysis. The current study aimed to explore how the different treatments used in the STOPS trial influence individual items of the ODI at different follow-up stages. Such an approach has been previously used to understand the mechanisms of how cognitive-behavioural therapy positively influenced insomnia and depression [33]. Given that the STOPS trial reported a significantly greater improvement in overall disability with individualized physiotherapy compared to advice [32], we hypothesised that the variable of treatment group would positively influence several items of the ODI. As an exploratory study, however, we did not impose any prior assumptions on the direction or magnitude of interactions between any items of the ODI.

line 99: is it [22] or [23] or both ?

Reply: The reference pertains to the primary results of the STOP trial, so it was reference 22 in the original manuscript. Reference 23 in the original manuscript referred to the protocol. 

The introduction ends abruptly without the research question, without a working hypothesis, without the objectives, etc. This should be added.

Reply: We have added a clearer study aim and working hypothesis to the Introduction (L135).

The current study aimed to explore how the different treatments used in the STOPS trial influence individual items of the ODI at different follow-up stages. Such an approach has been previously used to understand the mechanisms of how cognitive-behavioural therapy positively influenced insomnia and depression [33]. Given that the STOPS trial reported a significantly greater improvement in overall disability with individualized physiotherapy compared to advice [32], we hypothesised that the variable of treatment group would positively influence several items of the ODI. As an exploratory study, however, we did not impose any prior assumptions on the direction or magnitude of interactions between any items of the ODI.

Materials and Methods:

This part is the most difficult to understand and I think you are aware of it. This part is however very complete and I have nothing to say about the methodology which seems to be very rigorous. Much effort has been made to make it clear, but in my opinion there are still elements that need to be clarified to facilitate the reader's understanding.

Reply: We have tried to balance methodological rigor for statisticians who will scrutinize our methods, with ease of comprehension for readers unfamiliar with network analysis. We have made further changes to the Introduction (for more background on understanding network analysis) and Methods sections, for example as shown below: 

L99: 

In network analysis of the ODI for example, individual ODI items would be treated as nodes, and a network model would conceptualize LBP disability as a set of mutually interacting associations between these nodes. Associations between two nodes in a network are connected by an “edge” and reflect the magnitude of the relationship after statistically controlling for all other nodes in the network model [22]. Statistically, the association between two variables calculated in network analysis is analogous to the beta coefficient in a traditional multiple linear regression model, where one variable is the outcome and all the remaining variables are the predictors [22].

L222: 

Edges represent the existence of an association between two nodes, conditioned on all other nodes. Each edge in the network represents either a positive regularized association (blue edges) or a negative regularized association (red edges). Given their continuous nature, associations between items of the ODI reflect partial correlation coefficients, analogous to regression beta coefficients. Given that “group” was modelled with the advice group coded as 0 and the individualised physiotherapy group as 1, red edges (negative associations) between the “group” node and an ODI item would indicate that the individualised physiotherapy group had lower scores on that ODI item than the advice group [33]. If the two treatment groups had a similar influence on an ODI item score after controlling for all other items, an edge will not be present between the group and that ODI item. The thickness and colour saturation of an edge denotes its weight (the strength of the association between two nodes).

line 107 : how were patients diagnosed with LBP ?

Reply: We have added more details as to how patients were diagnosed with LBP. The information can be found in L153

Participants were eligible for the trial if they: had a primary complaint of LBP (pain between the inferior costal margin and inferior gluteal fold), with or without referred leg pain, between 6 weeks and 6 months duration, were aged 18-65 years, spoke English, and belonged to one of five low back disorder subgroups (disc herniation with associated radiculopathy, reducible discogenic pain, non-reducible discogenic pain, zygapophyseal joint dysfunction, and multifactorial persistent pain) [32]. Exclusion criteria were: the presence of a compensation claim, serious pathology (active cancer, cauda equine syndrome, foot drop making walking unsafe), pregnancy or childbirth within the last 6 months, history of lumbar spine surgery, spinal injections in the past six weeks, pain intensity < 2/10 (on a 0-10 numerical rating scale) or minimal activity limitation. All selection criteria, including the diagnosis of LBP, were confirmed by a physiotherapist after an initial 60-minute assessment before entry to the trial. 

line 113: how was the pain assessed at 2/10 (tool)

Reply: We have added the information in L161, which reads as:

pain intensity < 2/10 (on a 0-10 numerical rating scale)

line 138 : is it a typo problem [29] [29] [29] ?

Reply: We have corrected the typographical error in L203.

The data set was analysed with the R software (version 3.6.0, available at https://www.r-project.org) [40].

line 144-146:Why did you modify the ODI scale and replace the item on "sexual life" ? In my opinion, it should have been left for two reasons. First, this is an important question for LBP patients. And especially secondly because it modifies the scale that was originally validated with this item. It should be explained how changing the scale does not compromise its use in the study. Especially since the item "Work" which replaces it seems very important in the results ... it is in my opinion the first "problem" of the study.

Reply: We thank the Reviewer for these comments. In L215, we added a sentence to rationalize the reason for us selecting the previously modified version of the ODI, and that the modification has been shown to not adversely influence the construct validity of the questionnaire.

L215: 

The replacement of the “sex life” item with a “work/housework” item was previously undertaken to overcome frequent missing responses to the original sex life question [45]. This modified version has been shown to not adversely affect the construct validity of the ODI [44]. 

the second problematic point about the study is that I do not understand how we can individualize the IP and advice group ... I reread the materials and methods several times, I looked at the figures but I cannot understand how you draw conclusions by saying that the IP is better on an item compared to the advice group. This lack of understanding highlights the complexity of the study. This remark may surprise you but I think that the readers will not understand, by looking at the figures how you manage to differentiate the groups on the items.

It must be explained more clearly in the materials and method how the 2 groups will be differentiated. and in the discussion part, give examples to understand the difference between groups

Reply: We thank the Reviewer for these comments. We have addressed this by illustrating the meaning of an edge between two nodes in L222

Edges represent the existence of an association between two nodes, conditioned on all other nodes. Each edge in the network represents either a positive regularized association (blue edges) or a negative regularized association (red edges). Given their continuous nature, associations between items of the ODI reflect partial correlation coefficients, analogous to regression beta coefficients. Given that “group” was modelled with the advice group coded as 0 and the individualised physiotherapy group as 1, red edges (negative associations) between the “group” node and an ODI item would indicate that the individualised physiotherapy group had lower scores on that ODI item than the advice group [33]. If the two treatment groups had a similar influence on an ODI item score after controlling for all other items, an edge will not be present between the group and that ODI item. The thickness and colour saturation of an edge denotes its weight (the strength of the association between two nodes). 

Discussion:

The discussion is rather well structured and clear and I do not have any particular remarks. However, as I said, we do not understand the conclusions obtained with regard to the results.

Reply: We hope we have adequately addressed all comments raised by the Reviewer.

References: put the numbers in square brackets? [1], etc.

Reply: We have formatted the references in the bibliography consistently with the PLoS ONE author guidelines and the latest articles published on PLoS ONE (see https://doi.org/10.1371/journal.pone.0251513 below). Hence, we have retained the current formatting of the bibliography.

In conclusion, I would say that this study is very interesting, complete, well constructed and above all original. I recommend its publication AFTER making changes that will improve the reader's understanding.

Reply: We thank the Reviewer for these positive comments.

The 2 central points are:

- The problem of modifying the ODI scale on an item: explain how this does not call into question the validity of the scale.

- And above all: explain to the reader how to read the figures to highlight the differences between the 2 groups in the study. The use of example seems important to me since the target audience (physiotherapists) are not always clear about network analysis.

In any case, well done for this work.

Reply: We thank the Reviewer for all comments and we hope we have adequately addressed all of them in this Revision. 

 

Reviewer #3: 

1. Define LBP

Reply: Based on the response to Reviewer 2, we have now provided the definition as part of the description of the participants in L153.

Participants were eligible for the trial if they: had a primary complaint of LBP (pain between the inferior costal margin and inferior gluteal fold), with or without referred leg pain, between 6 weeks and 6 months duration, were aged 18-65 years, spoke English, and belonged to one of five low back disorder subgroups (disc herniation with associated radiculopathy, reducible discogenic pain, non-reducible discogenic pain, zygapophyseal joint dysfunction, and multifactorial persistent pain) [32]. Exclusion criteria were: the presence of a compensation claim, serious pathology (active cancer, cauda equine syndrome, foot drop making walking unsafe), pregnancy or childbirth within the last 6 months, history of lumbar spine surgery, spinal injections in the past six weeks, pain intensity < 2/10 (on a 0-10 numerical rating scale) or minimal activity limitation. All selection criteria, including the diagnosis of LBP, were confirmed by a physiotherapist after an initial 60-minute assessment before entry to the trial. 

2. please Specify the conditions/disease which causes LBP in your inclusion criteria

Reply: This information has been added to the description of participants (relevant section highlighted below): 

L153

Participants were eligible for the trial if they: had a primary complaint of LBP (pain between the inferior costal margin and inferior gluteal fold), with or without referred leg pain, between 6 weeks and 6 months duration, were aged 18-65 years, spoke English, and belonged to one of five low back disorder subgroups (disc herniation with associated radiculopathy, reducible discogenic pain, non-reducible discogenic pain, zygapophyseal joint dysfunction, and multifactorial persistent pain) [32].

3. why you have taken 18-65 age groups subjects.

Reply: This age category is a very common inclusion criterion in LBP clinical trials (Bråten et al., 2020; Vibe Fersum et al., 2019), reflecting the minimum age of consent up to the age of typical retirement in Australia. Older age groups would be more prone to non-specific and degenerative disorders inconsistent with the target population as defined above. 

4. mention gender distribution ( how many males and females)

Reply: We included information on the gender distribution in L171.

Participants were randomised via an offsite randomisation service to either individualised physiotherapy (n=156, 76 female and 80 male) or guideline-based advice (n=144, 71 female and 73 male). 

5. mention tool used for measuring pain intensity

Reply: In response to Reviewer 2, we provided this information in L161.

pain intensity < 2/10 (on a 0-10 numerical rating scale)

6. mention ODI psychometric properties

Reply: In response to Reviewer 2, we provided brief information of the ODI’s psychometric properties in L66 and L215.

A primary outcome measure used in LBP research is the Oswestry Disability Index (ODI) for measuring the impact of LBP on activities of daily living [9]. The ODI is composed of 10 items and the aggregate score indicates the overall disability level attributable to LBP [9, 10]. ODI has demonstrated good internal consistency [11],intrinsic validity [12], test-retest reliability [13], and responsiveness [13]. 

The replacement of the “sex life” item with a “work/housework” item was previously undertaken to overcome frequent missing responses to the original sex life question [45]. This modified version has been shown to not adversely affect the construct validity of the ODI [44]. We treated the items of the ODI as continuous variables and applied a nonparanormal transformation to ensure that these ten variables were multivariate normally distributed [46]. Higher ODI aggregate and item scores represent greater disability. 

7. duration of protocol should be mention per day tretment duration/days per week/ total weeks/days

Reply: In the original manuscript, we wrote:

Guideline-based advice comprised 2 x 30 minute sessions with a physiotherapist over a 10-week period based on the approach described by Indahl [29].

Individualised physiotherapy comprised 10 x 30-minute physiotherapy sessions over 10 weeks.

We have added extra information to this ection:

L177: 

Guideline-based advice comprised 2 x 30-minute sessions with a physiotherapist over 10 weeks based on the approach described by Indahl [35]. The first session was delivered shortly after randomisation, and the second approximately 4-5 weeks later. 

L184: 

Individualised physiotherapy comprised 10 x 30-minute physiotherapy sessions over 10 weeks. Sessions were typically spaced weekly, although therapists had the option to deliver treatment more frequently in the first 2-3 weeks and less frequently in the last 2-3 weeks if clinically indicated. 

8. Randomisation procedure ( GRoup division is not mentioned clearly- Group A/ Group B-so on)

Reply: We have included a section on the randomization process, in L165:

Randomisation

Eligible participants were randomised into one of two treatment groups via a computer-generated randomisation sequence; Individualised physiotherapy or advice. Allocation was concealed using an offsite randomisation service that allocated participants to treatment groups. 

9. review your Methodology ( intervention and data collection)

Reply: Several changes have been made to the reporting of our methodology, including information relating to participants, randomization, and interventions (described above). A new subheading specifically dedicated to data collection methods has been added (L197).

Data collection

Self-administered questionnaires containing the ODI among other outcomes were posted to participants at each time point. Non-respondents were followed up directly and via alternative provided contacts if necessary. 

10. check references format

Reply: We have checked our references to ensure that it follows the author guidelines of PLoS ONE. Please see a screenshot of the updated reference list below.

11. table 1 is not easily understandable. make it in simple format

Reply: We thank the Reviewer for this comment. Table 1 represents the exact wording of the ODI questionnaire used in the present study. The authors feel it is important to present it in its entirety, to maximize reproducibility. However, the original Table 1 does not need to be present in the main manuscript. We have moved this table to the supplementary and replaced it with an abbreviated version. 

12. review references and their format.

Reply: We have checked our references to ensure that it follows the author guidelines of PLoS ONE.

13. make a table of results obtained at different intervals

Reply: We have included tables of the edge weights variability and centrality values in the supplementary material. It would not be suitable to have results presented in the main manuscript as both figures and tables, due to information duplication and the large size of the data tables which are better suited to supplementary appendices. 

Edge weights variability and centrality indices values are reported in the manuscript graphically (Figures 2, 3), but also as tabular text in the supplementary material.

References

Bråten, L.C.H., Grøvle, L., Espeland, A., Pripp, A.H., Grotle, M., Helllum, C., . . . group, A.I.-s., 2020. Clinical effect modifiers of antibiotic treatment in patients with chronic low back pain and Modic changes - secondary analyses of a randomised, placebo-controlled trial (the AIM study). BMC Musculoskelet Disord 21, 458-.

Vibe Fersum, K., Smith, A., Kvale, A., Skouen, J.S., O'Sullivan, P., 2019. Cognitive functional therapy in patients with non-specific chronic low back pain-a randomized controlled trial 3-year follow-up. Eur J Pain.

---

## [Decision Letter · Decision Letter 1]

24 Jan 2022

Understanding how individualised physiotherapy or advice altered different elements of disability for people with low back pain using network analysis.

PONE-D-21-12519R1

Dear Dr. Liew,

We’re pleased to inform you that your manuscript has been judged scientifically suitable for publication and will be formally accepted for publication once it meets all outstanding technical requirements.

Kind regards,

Matias Noll, Ph.D

Academic Editor

PLOS ONE

Additional Editor Comments (optional):

Reviewers' comments:

Reviewer's Responses to Questions

**Comments to the Author**

1. If the authors have adequately addressed your comments raised in a previous round of review and you feel that this manuscript is now acceptable for publication, you may indicate that here to bypass the “Comments to the Author” section, enter your conflict of interest statement in the “Confidential to Editor” section, and submit your "Accept" recommendation.

Reviewer #1: All comments have been addressed

Reviewer #2: All comments have been addressed

2. Is the manuscript technically sound, and do the data support the conclusions?

Reviewer #1: Yes

Reviewer #2: Yes

3. Has the statistical analysis been performed appropriately and rigorously? 

Reviewer #1: Yes

Reviewer #2: Yes

4. Have the authors made all data underlying the findings in their manuscript fully available?

Reviewer #1: Yes

Reviewer #2: Yes

5. Is the manuscript presented in an intelligible fashion and written in standard English?

Reviewer #1: Yes

Reviewer #2: Yes

6. Review Comments to the Author

Reviewer #1: I am happy with the author's responses and have nothing further to add. this is an interesting paper

Reviewer #2: The authors have taken into account all the remarks. They responded with rigor and seriousness. In my opinion, the work is much more understandable! This was the major point. I therefore recommend the publication of this very interesting work!

7. PLOS authors have the option to publish the peer review history of their article (what does this mean?). If published, this will include your full peer review and any attached files.

Reviewer #1: No

Reviewer #2: **Yes: **Lucas Martinez

---

## [Editor Report · Acceptance letter]

2 Feb 2022

PONE-D-21-12519R1 

Understanding how individualised physiotherapy or advice altered different elements of disability for people with low back pain using network analysis. 

Dear Dr. Liew:

I'm pleased to inform you that your manuscript has been deemed suitable for publication in PLOS ONE. Congratulations! Your manuscript is now with our production department. 

Kind regards, 

on behalf of

Dr. Matias Noll 

Academic Editor

PLOS ONE